# Retrofitting metal-organic frameworks

Christian Schneider [1,3], David Bodesheim[1,3], Julian Keupp [2], Rochus Schmid [2] & Gregor Kieslich [1]*

The post-synthetic installation of linker molecules between open-metal sites (OMSs) and undercoordinated metal-nodes in a metal-organic framework (MOF) — retrofitting — has recently been discovered as a powerful tool to manipulate macroscopic properties such as the mechanical robustness and the thermal expansion behavior. So far, the choice of cross linkers (CLs) that are used in retrofitting experiments is based on qualitative considerations. Here, we present a low-cost computational framework that provides experimentalists with a tool for evaluating various CLs for retrofitting a given MOF system with OMSs. After applying our approach to the prototypical system CL@$Cu_3BTC_2$ (BTC = 1,3,5-benzentricarboxylate) the methodology was expanded to NOTT-100 and NOTT-101 MOFs, identifying several promising CLs for future CL@NOTT-100 and CL@NOTT-101 retrofitting experiments. The developed model is easily adaptable to other MOFs with OMSs and is set-up to be used by experimentalists, providing a guideline for the synthesis of new retrofitted MOFs with modified physicochemical properties.

[1] Department of Chemistry, Technical University of Munich, 85748 Garching, Germany. [2] Computational Materials Chemistry Group, Fakultät für Chemie und Biochemie, Ruhr-Universität Bochum, Bochum, Germany. [3]These authors contributed equally: Christian Schneider, David Bodesheim. *email: Gregor.Kieslich@tum.de

Metal-organic frameworks (MOFs) combine the variety of inorganic coordination chemistry with the large chemical parameter space of organic chemistry[1–3]. At the heart of MOFs is their modular building block principle, which provides experimentalists with the control of structure and functionality through linker topicity, linker chemistry and metal-node symmetry[4,5]. In turn, MOFs show a variety of fundamentally interesting and technically relevant properties. For instance, one of the recent breakthrough discoveries is the use of MOFs as working media in highly efficient, non-toxic water recovery technologies[6,7]. Other intriguing examples are the application of defect-engineered MOFs as Lewis-acid catalysts[8,9], the use of MOFs (as precursor) for the synthesis of cathode materials in the oxygen evolution reaction[10–12] and synthesis of electrically conducting systems with remaining guest-accessible porosities to name just a few[13,14]. In the pursuit to further optimize the physicochemical properties of MOFs for certain applications, post-synthetic modification (PSM) methods have proved to be an important tool[15–17]. For instance, a porous but catalytic inactive MOF can be transformed into a robust heterogeneous catalyst by post-synthetic immobilization of a catalytically active iron complex at a functional group of the linker[18]. Likewise, breathing behavior can be introduced in a formally non-flexible MOF through post-synthetic functionalization of the linker molecules[19], nicely highlighting the opportunities that come with PSM methods. Moreover, it has been shown that the properties of acid-gas degraded zeolitic imidazolate frameworks can be recovered by post-synthetic treatment with a fresh linker solution[20]. Looking at PSM for MOFs from a more fundamental perspective, such methods are only possible due to the combination of the underlying coordination chemistry, the available functionality of the organic back-bone and the guest-accessible porosity of MOFs, representing a unique toolbox for experimentalists to fine-tune physicochemical properties.

In the search for new PSM methods, Yaghi and co-workers have recently introduced 'retrofitting' as a useful and intuitive categorization[21]. In the most general definition, retrofitting of a MOF describes the post-synthetic installation of additional linkers between undercoordinated metal nodes or between open metal sites (OMSs) of a MOF[21]. Subsequently, retrofitting has been discovered as a powerful approach to manipulate macroscopic physicochemical properties such as the mechanical robustness, the thermal expansion behavior and responsivity of flexible MOFs to guest adsorption[21–23]. In a typical retrofitting experiment, a MOF with guest accessible OMSs or labile monotopic ligands is exposed to a molecule with at least two available coordination sites such as nitrile or carboxylate groups. This molecule then bridges two OMSs or two undercoordinated metal nodes, adding an additional connectivity between two different metal nodes. In the spirit of retrofitting which originally describes the addition of new components to an existing system, e.g., in construction to reinforce the structural stability of historical buildings[24,25], we refer to these guest molecules as *cross linkers* (CLs). It is important to note that in 2016 H. C. Zhou and co-workers demonstrated the possibility of installing two linear CLs with different lengths at two different positions in PCN-700 and referred to this approach as sequential linker installation (SLI);[26,27] however, since retrofitting as a concept overarches several different areas of on-going research and science[21,22,24,25,28,29], we strongly believe that retrofitting is the more intuitive and general categorization and is used throughout this manuscript.

Today, several intriguing examples are known where retrofitting was used to manipulate the physicochemical properties of a parent MOF. For example, O. Yaghi et al. used 4,4′-biphenyldicarboxylate as ditopic CL to render the Zr-based MOF-520 more robust towards high mechanical pressure[21]. In a similar fashion, Su and co-workers installed linear dicarboxylate CLs in a flexible Zr-based MOF to modulate the breathing behavior of the MOF and the sorption properties towards $N_2$ and $CO_2$[23]. In 2014, well before 'retrofitting' was introduced as a concept for MOFs, A. A. Talin et al. showed that the redox-active CL TCNQ (7,7,8,8-tetracyanoquinodimethane) can bridge two OMSs in $Cu_3BTC_2$ (BTC = 1,3,5-benzenetricarboxylate, HKUST-1), which was observed to come with an increase in the electrical conductivity of the material[30]. Using the same system, we recently demonstrated that retrofitting can be used as a tool to fine-tune the negative thermal expansion of $Cu_3BTC_2$, where an allover stiffening of the material is mainly responsible for reduced negative thermal expansion behavior as a function of TCNQ incorporation[22]. Looking at some general experimental considerations, the CL for a retrofitting experiment should be chosen to avoid post-synthetic linker exchange[17], i.e., the binding affinity of the metal-node with the CL should be weaker than with the linker. Additionally, when trying to rationalize the underlying mechanism of CL installation, diffusion limitations are expected to play a role, since diffusion pathways through the pores are successively blocked with increasing CL installation. While longer reaction times and relatively high temperatures can overcome such diffusion barriers[30,31], vapor phase infiltration at elevated temperatures with CLs that exhibit low sublimation temperatures have proved most suitable in this context[13,30,32]. It is also interesting to note that current literature examples are limited to retrofitting with ditopic CLs[21–23,26,27,31], with bulkier tritopic ligands further increasing concerns related to diffusion limitations. Lastly, it can be observed that retrofitting experiments mostly lead to a defective state of the CL@MOF system, with only a partial occupancy of the CL within the parent MOF network[21,22,31]. In the big picture, this situation draws a clear line between the defect chemistry of MOFs and purely inorganic materials, with MOFs exhibiting an additional degree of freedom for creating highly defective systems.

Despite the large number of MOFs with OMSs or labile monotopic ligands, the examples of retrofitted MOFs are currently limited to only a few systems. To fully understand and unravel the synthetic opportunities that come with retrofitting as PSM, more (systematic) retrofitted MOF series are required. When designing a retrofitting experiment in the lab, immediately the question arises, "how can we assess whether a cross linker (CL) is suitable for retrofitting of a given MOF?". D. S. Sholl and co-workers approached this question by applying a DFT-based screening study to identify Cu paddlewheel MOFs that feature OMSs for binding of TCNQ[33]. While this approach is expected to provide reliable results, a screening of different combinations of MOFs and CLs comes at a high computational cost. Likewise, the use of computational screening methods based on carefully parameterized force-fields requires experience with these computational methods, typically limiting the scope for experimentalists for exploiting such screening approaches. Consequently, a robust and adaptive approach is required which is both easy to use and computationally not demanding. Additionally, as such an approach is most valuable for experimentalists, the underlying methodology should reflect the experimentalists' chemical intuition and should be usable with a minimum of experience with computational methods.

In this work we present an easy-to-use low-cost computational framework that allows for evaluating the applicability of a certain CL for the use in a retrofitting experiment. We developed a program named RetroFit that is based on an open-source code and can guide experimentalists in the selection of suitable ditopic CLs for their retrofitting experiments. The approach is inspired by research efforts in the field of molecular modeling in the context of drug design, where molecular docking of small molecules to

macromolecular structures is predicted based on their geometry and interaction potential[34,35]. Whilst our approach is generally applicable to MOFs with OMS, we use $Cu_3BTC_2$ and a library of 20 CLs (16 different molecules, 4 of them allow for more than one distinct binding mode) with nitrile groups as a test system for RetroFit. After identifying suitable CLs, the practicability of the RetroFit algorithm is shown by synthesizing the most promising CL@$Cu_3BTC_2$ systems, obtaining a set of three new retrofitted CL@$Cu_3BTC_2$ systems with CL = TCNB, DCNB, and DCNT (TCNB = 1,2,4,5-tetracyanobenzol, DCNB = 1,2-dicyanobenzol and DCNT = 2,5-dicyanothiophene). Finally, we show that RetroFit can easily be extended to other MOF systems such as NOTT-100 and NOTT-101, where we identify several promising CLs for the synthesis of retrofitted systems with the general formula CL@NOTT-100 and CL@NOTT-101.

## Results and discussion

**The RetroFit algorithm.** Staying within the realm of ditopic CLs to bridge two OMSs via retrofitting, and raising the question which CLs are suitable for creating a CL@MOF system, two main requirements for a CL can be identified: (i) the presence of two suitable electron donating groups that can bind to the OMSs of the parent MOF and (ii) a size and geometry of the CL that brings these groups into a favorable vicinity of the OMSs for bond formation. Whereas (i) can easily be fulfilled by choosing CLs with two nitrile, amino or carboxylate groups, (ii) can be approached by analyzing the CL geometry compared to the spatial arrangement of the OMSs. Combining both requirements in a screening approach is expected to provide a good measure for estimating if the synthesis of such a CL@MOF system can lead to a stable host–guest system. Based on these considerations, we developed the RetroFit program code that allows for evaluating the applicability of CLs for the use in retrofitting experiments for a given MOF with two OMSs. RetroFit is designed to avoid expensive calculations and to circumvent time-consuming trial and error experiments, in total facilitating the discovery of new retrofitted CL@MOF systems. At the core of RetroFit is a minimization routine which optimizes the position of a given CL relative to the OMS positions. The spatial minimization is based on the binding energy of the CL to the OMSs as reflected in the model interaction potential (MIP). The MIP itself is used as input for RetroFit and is obtained via single point DFT calculations of a simple test system. A schematic of the required input and output of RetroFit is shown in Fig. 1, and a detailed workflow including a how-to guide is provided in the supplementary data files. Overviewing the workflow, the required input is (i) the energies of CL coordination to the specific OMS as a function of bond lengths and angles (i.e., the MIP) of the test system which is used in the minimization process, (ii) the structure of the MOF and in particular the spatial orientation of the OMSs represented in the parameters $R_{MM}$ and $\gamma$, and (iii) structural information about the CL that is tested which determines $R_{DD}$ and $\alpha$. The relative position of the two OMSs and the CL is defined by $R_{MD}$ and the two angles $\delta$ and $\theta$, as illustrated in Fig. 2. Since $\alpha$, $\gamma$, and $\theta$ are interdependent (see Supplementary Note 1), the geometry of the entire host guest system can be defined by a set of six parameters, i.e., ($R_{MM}$, $R_{DD}$, $R_{MD}$, $\alpha$, $\gamma$, and $\delta$) or ($R_{MM}$, $R_{DD}$, $R_{MD}$, $\alpha$, $\theta$, and $\delta$), or ($R_{MM}$, $R_{DD}$, $R_{MD}$, $\alpha$, $\gamma$, and $\theta$). The parameters $R_{MM}$, $\gamma$, $R_{DD}$ and $\alpha$ are extracted from the structure of the MOF and CL and treated as constants in our model, $\theta$ only depends on $\alpha$ and $\gamma$ and can be directly computed, whereas $R_{MD}$, $\delta$ are variables that are optimized during the RetroFit routine. Running through the RetroFit program for one given CL@MOF combination returns an energy penalty $\Delta E$ which compares the best orientation of the CL within the MOF system to the ideal binding distance and

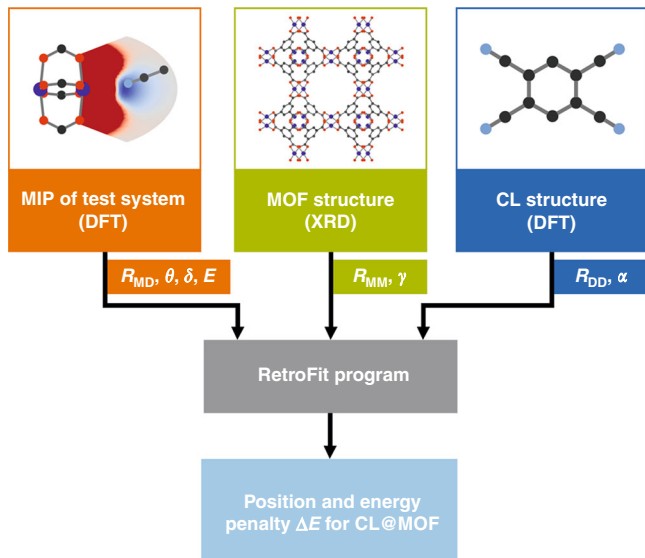

**Fig. 1** General workflow of the RetroFit program. Three sets of input data, i.e., the energies from single-point DFT calculations of a test system (orange), the relative position of the OMSs in the MOF (green) and the relative position of the donor groups of the CL (blue), are used by the RetroFit program (gray) to compute the optimal position of the CL within the MOF and the corresponding energy penalty (light blue). A workflow with more technical details and a how-to guideline are provided as a part of the Supplementary data files

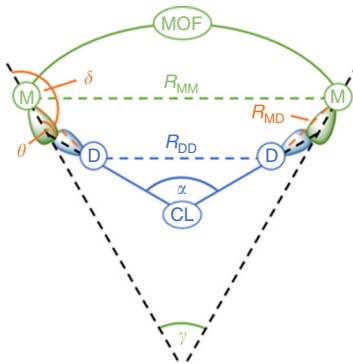

**Fig. 2** Geometric parameters of a general CL@MOF system. The parameters $R_{MM}$ and $\gamma$ are determined by the spatial arrangements of the OMSs of the parent MOF (green), while $R_{DD}$ and $\alpha$ are given by the CL (blue). $\theta$ is computed from $\alpha$ and $\gamma$, whereas the parameters $R_{MD}$ and $\delta$ (orange) are variables and optimized with respect to the underlying MIP. The two OMSs are related to each other by mirror symmetry

angle as provided by the MIP. The application of RetroFit to various CLs allows for ranking these after their energy penalties and further allows for grouping them into three categories (good fit, medium fit, poor fit – see Supplementary Note 7). This list can guide experimentalists in the identification and prioritization for the synthesis of promising CL@MOF systems. Furthermore, RetroFit allows to identify favorable spatial arrangements of the two electron donating groups (e.g., cis, ortho, etc.) from which further CL structures can be developed.

**RetroFit for CL@$Cu_3BTC_2$.** After introducing the general working principle of RetroFit, we applied RetroFit to the well-established $Cu_3BTC_2$ system. $Cu_3BTC_2$ was previously used in

retrofitting experiments to obtain TCNQ@Cu$_3$BTC$_2$ and TCNE@Cu$_3$BTC$_2$[13,22,32]. Notably, TCNQ and TCNE as prototypical CLs exhibit four nitrile groups that are available for the coordination to OMSs; however, due to the spatial orientation of the OMSs in Cu$_3$BTC$_2$ and size dependent restrictions given by the CL and MOF combination, only two nitrile groups coordinate to the MOF while the remaining two point to the center of the pore. Therefore, tetranitrile molecules such as TCNQ or TCNE are treated as bidentate CLs with two possible configurations within RetroFit. Looking at Cu$_3$BTC$_2$ in more detail, Cu$_3$BTC$_2$ crystallizes in the **tbo** topology and exhibits two types of large pores that in principle are accessible for guest molecules. In only one of these pores the Cu paddlewheels are oriented with the OMS pointing to the center of the pore to allow for retrofitting. In this pore, 12 OMSs on the axial positions of crystallographically equivalent Cu paddlewheel metal nodes point to the inside of the pore. The chemistry of such a Cu OMS shows a large propensity for the formation of Cu-N bonds, and in turn CLs with nitrile or amino groups are expected to be suitable CL candidates[36–38]. Following the workflow in Fig. 1, a MIP for the system is required which is the basis for the calculation of the aforementioned energy penalty $\Delta E$ for a given CL@MOF system. For Cu$_3$BTC$_2$ the retrofitting experiment is meant to bridge two of the 12 available copper OMS which are related to each other by a mirror plane. Therefore, it is suitable to focus on the MIP of one Cu OMS and subsequently using the mirror symmetry to obtain the interaction potential of the whole system (see Fig. 2). In Cu$_3$BTC$_2$ the distance between the Cu OMSs ($R_{CuCu}$) was extracted from crystallographic data[39]. The MIP input was obtained by single point DFT calculations (B3LYP/TZVPP level of theory, see Methods section for details) for a simple Cu(II)formate – acetonitrile complex (CH$_3$CN@Cu$_2$(HCOO)$_4$), where the metal to nitrile group distance ($R_{CuN}$), $\delta$ and $\theta$ of acetonitrile were varied. A subsequent interpolation of the 3 N dimensional data yielded the continuous MIP surface that represents one side of the CL-OMS interaction within the MOF, see Fig. 1 orange box for a representation of the MIP with fixed $\theta$ angle and Supplementary Note 6 for an extensive error analysis of the interpolation. In CH$_3$CN@Cu$_2$(HCOO)$_4$ the global energy minimum is reached for a linear coordination of acetonitrile ($\delta = \theta = 180°$) and a distance of $R_{CuN} = 2.2$ Å which we define as zero, i.e., $\Delta E = 0$. In the following, all energies for the CL@Cu$_3$BTC$_2$ system are expressed as energy difference $\Delta E$ from this ideal configuration. Lastly, the geometries of the different CLs are required, which were obtained by DFT optimization using the B3LYP hybrid functional and a 6–31 G basis set[40,41], see Methods section for details. The input of 20 dinitrile-CLs as *.xyz-files are given in the supplementary data files. RetroFit applied to one CL@Cu$_3$BTC$_2$ system varies the spatial orientation of the CL with respect to the OMSs within the MOF to minimize $\Delta E$ through the parameters $R_{CuN}$, $\delta$ and $\theta$. In turn, the application of RetroFit to various CLs allows to rank these after their energy penalties. Based on this methodology, we have applied RetroFit to the 20 different CLs$_{32}$ which are shown in Fig. 3.

To cover a broad range of possible CLs with various geometries we tested acyclic (e.g., **7**, and **10**) and (hetero-)cyclic systems with 3-, 4-, 5-, and 6-membered rings (e.g., **18**, **15**, **9**, and **5**), where the nitrile groups are in geminal, cis (e.g., **1**), ortho (e.g., **3**) or meta (e.g., **19**) positions, see Fig. 3. By using RetroFit, $\Delta E$ for the 20 tested CL@Cu$_3$BTC$_2$ combinations were calculated and are given in the energy map shown in Fig. 4 and the bar plot (see below). The global minimum of the energy map $\Delta E_{min}$, i.e., the molecule parameters that suggest the ideal fit, is located at $R_{NN} = 5.81$ Å and $\alpha = 60°$. Interestingly, the dependence of $\Delta E$ on the distance between the nitrile groups within the CL ($R_{NN}$) is significantly stronger than on $\alpha$. Therefore, molecules with similar $R_{NN}$

distances fit comparably well such as TCNQ (**13**, $R_{NN} = 4.447$ Å, $\Delta E = 2.24$ kcal mol$^{-1}$), TCNE (**2**, $R_{NN} = 4.436$ Å, $\Delta E = 2.29$ kcal mol$^{-1}$) and malononitrile (**10**, $R_{NN} = 4.431$ Å, $\Delta E = 2.32$ kcal mol$^{-1}$). Looking for trends, nitriles in ortho positions of small cyclic systems as given in tetracyanobutadiene (**16**) as well as geminal nitriles as available in TCNQ show a good fit with Cu$_3$BTC$_2$, while $\Delta E$ increases for larger ring systems and nitrile groups in cis position. In contrast, the meta configuration in CLs with 6-membered rings (**20**, $R_{NN} = 6.893$ Å for TCNB) shows a relatively high energy penalty of $\Delta E = 4.815$ kcal mol$^{-1}$ which is therefore less favorable for retrofitting Cu$_3$BTC$_2$ (see discussion in Supplementary Note 7). TCNQ as the prototype CL has a relatively low value of $\Delta E = 2.24$ kcal mol$^{-1}$, which is in good agreement with the results of previous studies[30,42,43]. Comparing TCNQ to other tested CLs, only CLs based on 3-, and 4-membered ring systems show smaller $\Delta E$ values. Such molecules often show flashpoints below 150 °C and are therefore less suitable for retrofitting experiments which involves vapor phase infiltration at temperatures typically above 100 °C. In order to evaluate the accuracy of the RetroFit routine, the results are compared with existing computational data from the literature on TCNQ@Cu$_3$BTC$_2$. A. A. Talin et al. modeled the binding of TCNQ in Cu$_3$BTC$_2$ by using DFT (UB3LYP[41,44]/VTZP level of theory[45]) and obtained a geometry optimized crystal structure of CL@Cu$_3$BTC$_2$ with CL = TCNQ[30]. Using their results as a benchmark and comparing these with the output values from RetroFit, we indeed observe a good correlation which is summarized in Supplementary Table 5. The largest deviations occur due to CL deformations, which are not accounted for in our model. For instance, $R_{NN}$ of TCNQ in the literature structure of TCNQ@Cu$_3$BTC$_2$ is increased by 0.167 Å to $R_{NN}' = 4.614$ Å compared to the free molecule, which results in a $R_{CuN}$ distance which is 0.168 Å shorter and in turn leads to a stronger coordination bond. Taking the distortion of TCNQ into account by using $R_{NN}'$ and $\alpha'$ from the literature as input for RetroFit, a significantly lower energy penalty of $\Delta E = 1.59$ kcal mol$^{-1}$ is obtained (see Supplementary Table 5). This example shows that despite the heuristic approach of RetroFit, the binding situation is reasonably represented. It seems, however, that RetroFit slightly overestimates the energy penalty, whereas in the real CL@Cu$_3$BTC$_2$ system structural deformations can occur that further minimize the total energy of the CL@MOF system, i.e., a slight increase in energy due to unfavorable molecule deformation can be overcome through the formation of stronger coordination bonds. Therefore, CLs with low energy penalties are expected to lead to a stable CL@Cu$_3$BTC$_2$ system and structural deformations can be expected to only play a minor role. Similarly, CL@Cu$_3$BTC$_2$ systems with relatively large energy penalties can be ruled out, whilst assumptions drawn for CL@MOF combinations with intermediate energy penalties are more inconclusive. For our CL@Cu$_3$BTC$_2$ system, we observe that CL = TCNB (**3**), DCNB (**5**) and DCNT (**6**) are promising candidates. These combine low to medium $\Delta E$ values with experimental applicability given by the physical properties of the CLs, and the synthesis of the corresponding CL@Cu$_3$BTC$_2$ systems is described in the next paragraph.

**Proof of concept – experimental validation.** After having identified several promising CLs for retrofitting Cu$_3$BTC$_2$, the synthesis of new CL@Cu$_3$BTC$_2$ materials was attempted. Following the results discussed in the previous section, we selected 1,2-dicyanobenzene (**5**, DCNB), 1,2,4,5-tetracyanobenzene (**3**, TCNB) and 3,4-dicyanothiophene (**6**, DCNT) as promising CLs for proof-of-principle experiments. These CLs show low and intermediate $\Delta E$ values, are commercially available and easy to handle. All of these

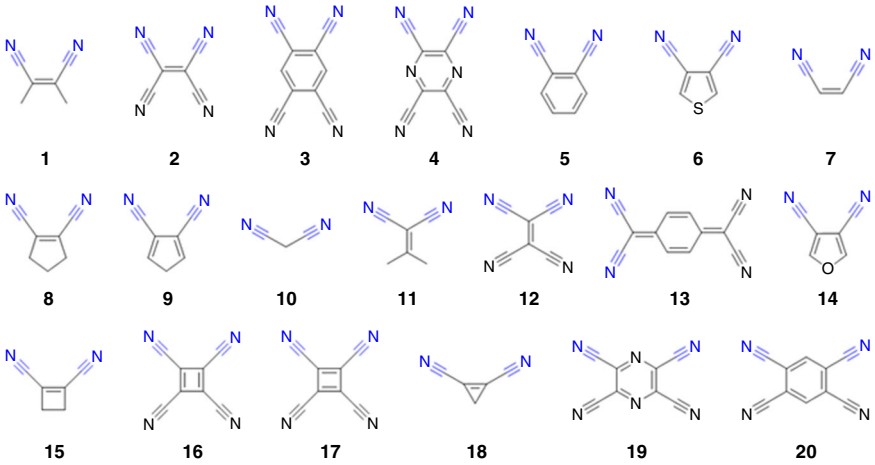

**Fig. 3** Library of potential CLs evaluated with RetroFit. Structural parameters were obtained from geometry optimized structures in the gas phase. The coordinates can be found in the program package provided in the Supplementary data files

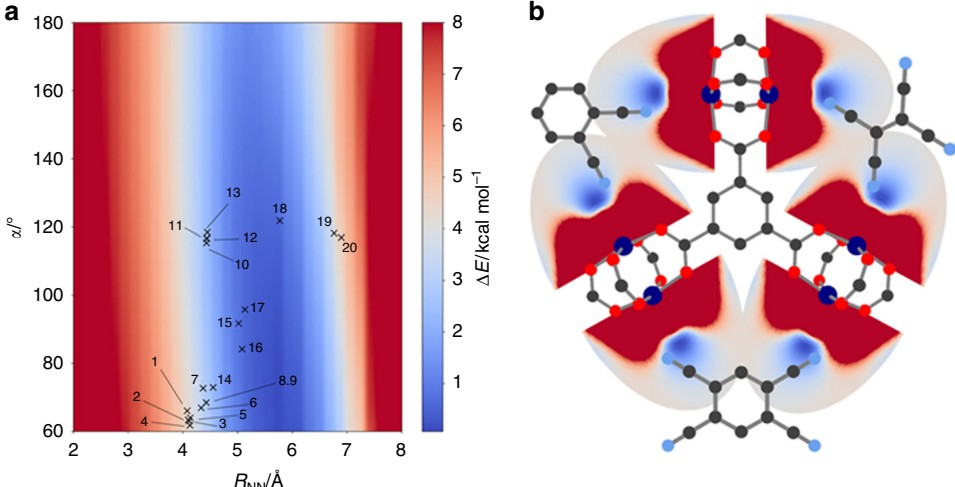

**Fig. 4** RetroFit results for $Cu_3BTC_2$. **a** RetroFit map for $Cu_3BTC_2$ and the tested dinitrile-CLs. The energy penalty $\Delta E$ for the CL parameters $R_{NN}$ and $\alpha$ is given as color code and increases from blue to red. Values of $\Delta E$, which exceed the color scale bar are set to 8 kcal mol$^{-1}$ for better visualization. The smallest $\Delta E$ and in turn the best fit is obtained for CLs with 3 and 4-membered rings (**15**, **16**, **17** and **18**). A table overviewing the results is given in Supplementary Note 7 (Supplementary Table 3). **b** Visualization of the fit of the CLs **5**, **12**, and **20** with ortho, geminal and meta configuration, respectively. Cu, C, N, and O atoms are shown in dark blue, black, blue and red. The potential around the OMSs is illustrated as $\Delta E$ increasing from blue to red

molecules show sublimation temperatures feasible for vapor phase loading experiments that currently seem to be the preferable experimental pathway[13,22]. Following the established solvent-free guest loading procedure[13], we prepared CL@$Cu_3BTC_2$ compounds with one molar equivalent of DCNB, TCNB and DCNT per $Cu_3BTC_2$. Such a loading situation corresponds to two CLs per large pore, which has been previously determined as the maximum loading for TCNQ[13,46]. Details on the synthetic procedure and characterization techniques are provided in the Methods section (see below) and in the references 13 and 22. After infiltration of $Cu_3BTC_2$ with the various CLs, powder X-ray diffraction (PXRD) confirms that the crystallinity of all samples is maintained (Fig. 5). An increased intensity of the (111) reflection is observed in all cases, which has previously been attributed to the chemisorption of guest molecules to two neighboring Cu paddlewheels within the (111) plane[13]. Closer investigation of the PXRD pattern reveals new reflections which are weak in intensity, see inset in Fig. 5. These new reflections are forbidden in the face-centered cubic space group of $Cu_3BTC_2$ ($Fm$-3m) and have been previously observed for TCNQ@$Cu_3BTC_2$[13]. Indexing of the PXRD pattern

suggests a primitive space-group, such as $Pn$-3m, pointing at some sort of ordering of the guest molecules within the two chemically different pores of $Cu_3BTC_2$. Notably, other symmetry-reduction pathways are possible, but when staying within the $3\sigma$ criterion, the Pawley fit with $Pn$-3m produces the most reasonable fit in all cases. Attempts of structure solution are challenged by the expected partial occupancy and complicated ordering mechanism of CL molecules within the pores and have not been successful so far. Additional proof for the presence of the CLs are new signals in the IR spectra that are related to the guest molecules, even though the nitrile bands are less intense when compared to TCNQ@$Cu_3BTC_2$ (Supplementary Note 10). To further underpin the accommodation of the CLs inside $Cu_3BTC_2$, nitrogen sorption isotherms (Fig. 6) and scanning electron microscopy images were recorded (Supplementary Note 11). From the initial slope of the type I isotherms comparable BET surface areas of 1170.3 m$^2$ g$^{-1}$ (DCNT), 1151.9 m$^2$ g$^{-1}$ (DCNB) and 1135.2 m$^2$ g$^{-1}$ (TCNB) were determined. These values are significantly lower than for pristine $Cu_3BTC_2$ (1873.8 m$^2$ g$^{-1}$) but higher than for TCNQ@$Cu_3BTC_2$ (869.9 m$^2$ g$^{-1}$), which is ascribed to the smaller size of DCNT,

DCNB and TCNB compared to TCNQ. For all new CL@$Cu_3BTC_2$ powders, SEM images confirm the absence of unreacted CLs or the formation of amorphous byproducts (see Supplementary Note 11). This is different when compared to the case of TCNQ@$Cu_3BTC_2$, where the formation of CuTCNQ nano-wires on the surface of the MOF crystals was observed[13]. Comparing the electrical conductivities of DNCT@$Cu_3BTC_2$, DCNB@$Cu_3BTC_2$

and TCNB@$Cu_3BTC_2$, all materials are electrical insulators (see Supplementary Note 12). This might be a result of different redox potentials of the CLs and the presence of the by-phase in TCNQ@$Cu_3BTC_2$ (a more detailed discussion can be found in Supplementary Note 12). Lastly, we would like to note that tetracyanoethylene (TCNE), which shows an excellent fit according to RetroFit, was already employed by D'Alessandro and coworkers;[32] however, our attempts of the synthesis of TCNE@-$Cu_3BTC_2$ led to a unexpected PXRD pattern that might either point to a transformation or a relatively drastic change of structure which we could not yet identify (see Supplementary Fig. 19). In summary, following the results from RetroFit, we successfully synthesized and characterized three new retrofitted CL@$Cu_3BTC_2$ systems namely DCNT@$Cu_3BTC_2$, DCNB@$Cu_3BTC_2$ and TCNB@$Cu_3BTC_2$, highlighting the applicability of the RetroFit algorithm.

**Beyond $Cu_3BTC_2$.** To show that RetroFit is easily transferable to other CL@MOF systems, we used RetroFit to screen various dinitrile-CLs for retrofitting NOTT-100 ($Cu_2$(BPTC), BPTC = biphenyl-3,3′,5,5′-tetracarboxylate) and NOTT-101 ($Cu_2$(TPTC), TPTC = [1,1′:4′,1″]terphenyl-3,3″,5,5″-tetracarboxylate) MOFs[47]. Both MOFs crystallize in the **nbo** topology and are built from Cu paddlewheel nodes and tetratopic rectangular-shaped linkers. Since NOTT-100 and NOTT-101 have Cu-based OMSs similarly to $Cu_3BTC_2$, the same MIP can be used as input for RetroFit. In contrast to $Cu_3BTC_2$, the crystal structures of both NOTT systems exhibit two crystallographically distinct binding sites. We refer to these sites and corresponding $R_{CuCu}$ and γ values as 3,5-position and 3,3′- (NOTT-100) or 3,3″-positions (NOTT-101) which is adapted from the nomenclature of the linker molecules (see Fig. 7 for a visualization of this situation). In turn, we can generate two sets of $\Delta E$ values, which rank the different CLs after the two different binding situations. The obtained $\Delta E$ values are listed in Fig. 8 and compared with $Cu_3BTC_2$. For the binding site in the 3,5-position, the RetroFit maps (see Supplementary Note 8) resemble the situation previously obtained for $Cu_3BTC_2$ (Fig. 4) due to the similar geometry; however, significant differences can be observed for the 3,3′ and 3,3″ binding site. It should be noted that in both NOTT systems, the Cu OMSs in 3,3′ (or 3,3″) position exhibit a small torsion angle. In RetroFit, this torsion angle is neglected for all CLs as the routine translates a 3D multiparameter problem into 2D space. Presumably, this adds another offset to the $\Delta E$ ranking; however, as a pronounced effect of the $R_{NN}$ distance has been found for $Cu_3BTC_2$, the torsion angle is expected to play only a minor role. The results of

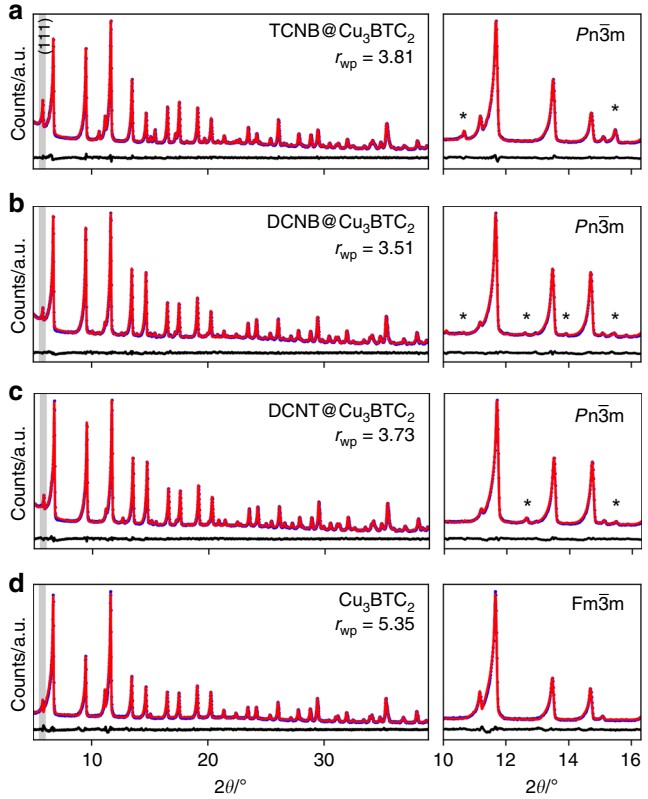

**Fig. 5** PXRD data and Pawley profile fits for CL@$Cu_3BTC_2$. Pristine $Cu_3BTC_2$ **d** crystallizes in the face-centered cubic space group *F*m-3m. Upon retrofitting with CL = TCNB **a**, DCNB **b**, or DCNT **c**, new reflections occur (zoom in) which can be accounted for by using the primitive space-group *P*n-3m. Experimental data, Pawley profile fits and difference curves are shown in blue, red, and black, respectively. The $r_{wp}$ fitting parameters are provided in the top right corner, respectively. The (111) reflection at 5.8° is indicated in gray

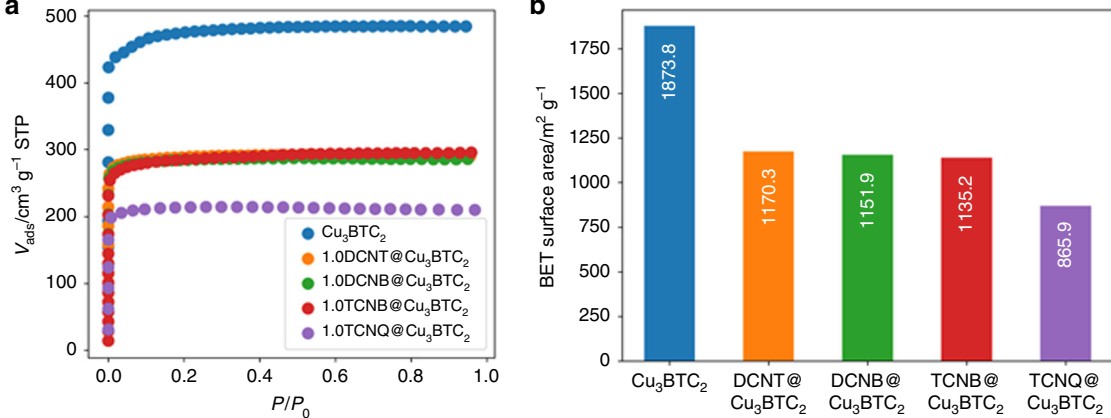

**Fig. 6** Porosimetry measurements of CL@$Cu_3BTC_2$. Nitrogen adsorption isotherms **a** and calculated BET surface areas **b** for samples with CL = DCNT (orange), DCNB (green), TCNB (red), and pristine $Cu_3BTC_2$ (blue). All new materials show a high residual porosity after the retrofitting experiment

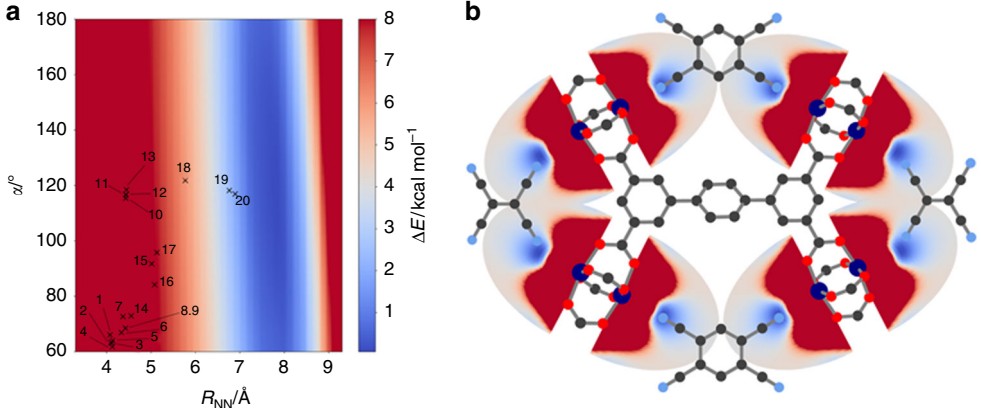

**Fig. 7** RetroFit results for NOTT-101. **a** RetroFit map for the 3,3″-position and the CL library (Fig. 3). Values of $\Delta E$, which exceed the color scale bar are set to 8 kcal mol$^{-1}$ for better visualization. The lowest $\Delta E$ for the 3,3″-position is obtained for CL **20** (TCNB, meta position), although CLs with even larger $R_{NN}$ distances will show better fit. **b** Visualization of the implications for retrofitting NOTT-101. It is conceivable that two different CLs can be installed in NOTT-101, e.g., TCNB in the 3,3″-position and TCNE in the 3,5-position. Cu, C, N, and O atoms are shown in dark blue, black, blue, and red. The potential around the OMSs is illustrated as $\Delta E$ increasing from blue to red

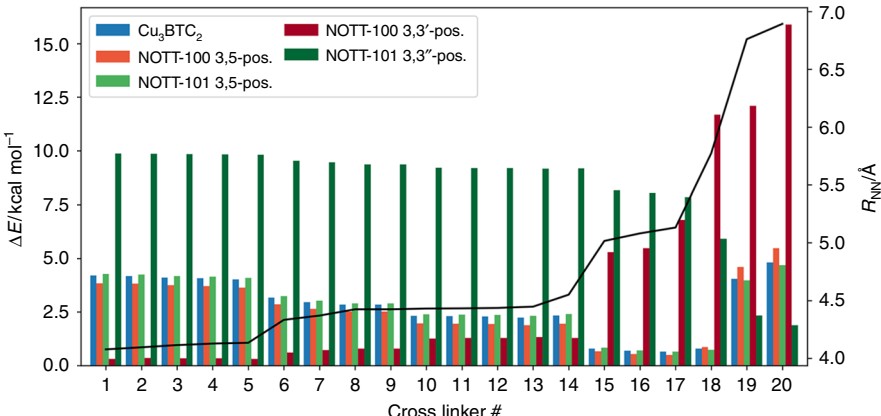

**Fig. 8** Summary of the RetroFit results. The energy penalties $\Delta E$ (bars) of all CL@MOF materials studied in this work are compared. The $R_{NN}$ distance of the CLs is visualized by the black line. Due to the anisotropy of the linker for the NOTT systems, two different positions, here referred to as 3,5 and 3,3′ (or 3,3″) are possible and have been tested. For bridging two OMSs in a MOF, distinct $R_{NN}$ windows that lead to low $\Delta E$ can be identified. Cu$_3$BTC$_2$, NOTT-100 (3,5-position) and NOTT-101 (3,5-position) have very similar $\Delta E$ values with an optimum at $R_{NN} \approx 5.1$ Å. NOTT-100 (3,3′-position) shows the best fit for low $R_{NN}$ values, whereas NOTT-101 (3,3″-position) shows the opposite trend

applying RetroFit to NOTT-100 and NOTT-101 are shown in Fig. 8 and compared to Cu$_3$BTC$_2$. In the 3,3′-position of NOTT-100, the short $R_{CuCu}$ distance of 7.601 Å in combination with the obtuse angle of $\gamma = 145.65°$ requires molecules with very short $R_{NN}$ distances (see Supplementary Fig. 16). The isoreticular expansion from NOTT-100 to NOTT-101 results in an increase of $R_{CuCu}$ for the 3,3′ (respectively 3,3″) position to 11.607 Å. Therefore, CLs with larger $R_{NN}$ distances, such as TCNB (**20**, meta), can be accommodated in the structure of NOTT-101 which do not fit into the structure of NOTT-100 or Cu$_3$BTC$_2$. These results suggest that a system with two distinctly different sites such as NOTT-101 can be fitted with two types of CLs, e.g., TCNE (**12**) or malononitrile (**10**) in the 3,5-position and a TCNB (**20**, meta) across the long side of the terphenyl-linker (3,3″-position), providing diffusion limitations can be minimized. Notably, TCNB is a candidate CL to bridge both the 3,5-position (ortho coordination) and the 3,3″ position (meta coordination), making it a particular interesting CL for future retrofitting experiments.

In conclusion we developed a low-cost computational framework called RetroFit which provides experimentalists with a tool to identify promising CLs for retrofitting MOFs. In a proof-of-principle study, we used RetroFit for screening various CLs for retrofitting experiments of the iconic Cu$_3$BTC$_2$ system. The synthesis of the most promising CL@Cu$_3$BTC$_2$ candidates was approached in the laboratory, obtaining the three new retrofitted systems DNCT@Cu$_3$BTC$_2$, DCNB@Cu$_3$BTC$_2$ and TCNB@Cu$_3$BTC$_2$. To show the applicability to other systems, we applied RetroFit to the well-known NOTT-100 and NOTT-101 MOFs, testing both possible bidentate bridging modes in these systems and identifying a few promising CL@NOTT-101 and CL@NOTT-100 candidates. We would like to note that RetroFit is not aiming at a quantitative description of the possible CL@MOF systems but is designed as an easy-to-use screening tool which can be operated without a profound background in computational chemistry. A detailed how-to guide including the program code and input data for reproducing the herein presented results are given in the supplementary data files. Looking at RetroFit from a more general perspective, the routine translates a complex multiparameter 3D optimization problem to a 2D model in which only the strongest chemical interactions, i.e., the metal-coordination bonds are accounted for. With the currently available MIP for nitrile-CLs and Cu-based OMS, RetroFit can easily be applied

to all MOFs with Cu-paddlewheel motifs as OMS, such as NU-111 (Cu$_3$(5,5',5"-(benzene-1,3,5-triyltris(buta-1,3-diyne-4,1-diyl))triisophthalate))[48], PCN-14 (Cu$_2$(5,5'-(anthracene-9,10-diyl)diisophthalate))[49] and NOTT-115 (Cu$_3$(4',4"',4""'-nitrilo-tris(([1"",1""'-biphenyl]-3,5-dicarboxylate))))[50] to name just a few. The next step in the development of RetroFit is the creation of a library of various MIPs for different metal nodes going beyond Cu-based OMS. Likewise, currently 20 dinitrile-based CLs are contained in the database, which can be expanded to amino-based or carboxylate-based CLs in the future.

Finally, it is important to emphasize a few additional points related to the retrofitting concept. Retrofitting as relatively new categorization of PSM is an intriguing approach for accessing the structural dynamics of MOFs. Today only a limited number of retrofitted MOFs exist, a situation that makes it difficult to oversee the full potential of the concept. Most notably, retrofitting as science overarching concept is not limited to MOFs. Even in the world of covalent organic frameworks (COFs) examples exist that can be classified under the umbrella of retrofitting, i.e., when metal complexes are used to cross-link two-dimensional COF sheets or when post-synthetic modification of functional groups leads to additional framework connectivity[28,29]. The future progress of the retrofitting concept is linked to the discovery of additional CL@MOF systems, which allows to answer open scientific questions on potential order phenomena of CLs within the parent MOFs as well as on the influence of CL size and CL-to-OMS bond strength on the macroscopic properties. With the synthesis of more retrofitted MOF systems, these questions can be approached, to which our RetroFit algorithm adds a powerful tool. Likewise, the concept itself can be extended. For instance, the incorporation of asymmetric CLs such as amino acids, or CLs that contain functional backbones such as optically active groups seem to be intriguing research directions. In these scenarios, diffusion limitations might start to play a more significant role but are difficult to estimate without experimental validation. The use of less strong donor CLs might be a solution to this problem, enabling the equilibrium to be reached at elevated temperatures. Therefore, we strongly believe the retrofitting concept has much to offer in the future, and that RetroFit can facilitate the discovery of new CL@MOFs, leaving the territory of trial and error experiments.

## Methods

**RetroFit algorithm**. RetroFit was written in the open source programming language Python™ (available at http://www.python.org) and was tested for versions 2.7 and 3.7. The algorithm uses three data sets (see Fig. 1) as input, i.e., the relative position of the OMSs within the MOF, the structural information of the CL, and a MIP representing the interaction between OMS and CL. Whereas the former is extracted from the MOF crystal structure, the geometry of the CL and the MIP are obtained by DFT calculations (see below). In the current version of RetroFit, the structural relation of the OMSs are extracted manually from the MOF crystal structure, whereas the geometry of the different CLs is provided as a library of molecular structure files (xyz-files). The continuous MIP is generated by interpolation of the input energies from single point DFT calculations of the model system. A detailed description of the algorithm is provided in Supplementary Note 3 and the workflow is shown in Supplementary Fig. 10. It should be noted that herein we release version 1.0 of RetroFit. Future developments of the code as well as a library of CLs and MIPs will be deposited on GitHub (https://github.com/GKieslich/RetroFit). We believe that the development of RetroFit is a task for computational scientists or users with advanced expertise on computation. On the other hand, RetroFit is designed to be applied by experimentalists with a minimum of experience with computational methods. Therefore, future developments of the program will be aiming on both to increase the capabilities of the program and to make RetroFit as user-friendly as possible.

**DFT calculations**. All molecule geometries were optimized with the Gaussian09 program package. Optimization was performed with DFT with a B3LYP hybrid functional and a 6–31 G basis set[40,41]. For the optimization the 'tight' convergence criterion was used and the Hessian was recalculated after each optimization step. All molecules were symmetry restricted during the optimization process (CLs **2**, **3**, **4**, **12**, **13**, **16**, **17**, **19**, and **20** were restricted to D$_{2h}$ symmetry while CLs **1**, **5–11**, **14**, **15**, and **18** were restricted to C$_{2v}$). The resulting Gaussian output file was converted

to the xyz-format using Open Babel (version 2.3.2) and then imported into the RetroFit tool using the Atomic Simulation Environment (ASE) (version 3.16.0) to compute R$_{NN}$ and α.

To access energies in the host–guest system consisting of a Cu paddlewheel-based MOF and a nitrile-CL, we optimized a Cu(II) formate paddlewheel and an acetonitrile molecule, respectively, and then arranged the two entities that the nitrile group points towards the OMS of the paddlewheel (CH$_3$CN@Cu$_2$(HCOO)$_4$). By varying R$_{CuN}$, δ and θ according to the parameters given in Supplementary Table 2, we obtained an energy of the system for every combination of the three parameters, which allows the generation of a MIP via linear interpolation. In general, the interpolation errors can be improved by a smaller step size but for the accuracy demands herein, the chosen step sizes are sufficient (see Supplementary Note 6). The configuration with the lowest energy is defined as 0 kcal mol$^{-1}$ and all energies are given as energy differences ΔE. Further, a factor of 2 is applied as the host–guest complex involves two coordination bonds of nitrile groups to two OMSs. The single point calculations were done on a DFT level of theory using the TURBOMOLE (V7.1) software package[51]. The hybrid functional B3LYP[41,44] was used with a TZVPP basis set[45] and a fine 'm5' grid[52] for all elements. The multipole-accelerated[53] resolution of the identity approximation[54,55] was used for performance reasons. Grimmes D3[56] was employed to properly account for dispersive interactions. The SCF convergence criterion was set to 10$^{-6}$ Hartree. Additional details on the single point DFT calculations can be found in Supplementary Note 2.

**Synthesis**. Cu$_3$BTC$_2$ was synthesized following the literature procedure[57]. The crystalline as-synthesized powder was solvent exchanged and activated following our previously published protocols[13]. The synthesis of CL@Cu$_3$BTC$_2$ with CL = 1,2-dicyanobenzene (DCNB), 1,2,4,5-tetracyanobenzene (TCNB), and 3,4-dicya-nothiophene (DCNT) was performed analogous to the reported procedure for TCNQ@Cu$_3$BTC$_2$[13]. Therefore, activated Cu$_3$BTC$_2$ (100 mg) and stoichiometric amounts of DCNB, TCNB, or DCNT were thoroughly mixed to yield mixtures with $x = n$(CL)/$n$(Cu$_3$BTC$_2$) = 1.0. The mixtures were filled into a glass ampule, which then was evacuated (10$^{-3}$ mbar) and flame-sealed. The ampules were placed in a convection oven at 180 °C for 72 h. After cooling down, the ampules were opened inside an Ar-filled glovebox and the CL@Cu$_3$BTC$_2$ powders were stored until further characterization. For all analytic methods, the powders were handled under inert conditions (Ar or vacuum) to avoid contamination from the atmosphere.

**Powder X-ray diffraction**. CL@Cu$_3$BTC$_2$ samples were filled into 0.7 mm borosilicate capillaries and sealed. The capillary was mounted onto a PANalytical Empyrean X-ray diffractometer operated in capillary mode using Cu Kα radiation, a focusing beam mirror with a 1/8° slit and 0.02 rad Soller slits as the incident beam optics and a 1/8° anti-scatter slit with 0.02 rad Soller slits and a Ni filter on the diffracted beam side. Diffraction data in the 2θ range of 5–50° with a step size of 0.013° was collected using a PIXcel 1D detector in scanning line mode. Quantitative data analysis (Pawley fits) was performed using the Topas Pro v5 software. The fitting parameter, i.e., weighted-profile R-factor (r$_{wp}$), is provided in the respective figures.

**Porosimetry measurements**. Inside of an Ar glovebox, ~60 mg of a sample were filled into a glass tube and evacuated at room temperature for 3 h at ~10$^{-5}$ mbar. The exact sample mass was determined and the isotherm was recorded on a Micromeritics 3flex at 77 K in the pressure range between 10$^{-3}$ and 10$^3$ mbar. The BET surface area was calculated from the initial slope (0.01 to 0.1 $P/P_0$) of the isotherm.

## Data availability
The authors declare that all data supporting the findings of this study are available within the article and its Supplementary Information or from the corresponding author upon reasonable request.

## Code availability
The RetroFit code including all required input files and a How-to-guide is freely available on GitHub, see https://github.com/GKieslich/RetroFit and licensed under a MIT license.

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

## Acknowledgements

We would like to thank Roland A. Fischer and Christoph Schran for insightful discussions. Mark D. Allendorf and A. Alec Talin are acknowledged for the support with electrical conductivity measurements and Pia Vervoorts and Dardan Ukaj for porosimetry measurements. C.S. acknowledges his scholarships from the Studienstiftung des Deutschen Volkes and from the Fonds der Chemischen Industrie (FCI). G.K. gratefully thanks the FCI and the DFG SPP1928 (COORNETs) for financial support.

## Author contributions

C.S., D.B. and G.K. developed the concept and algorithm for RetroFit. C.S. synthesized and characterized the new $CL@Cu_3BTC_2$ materials. J.K. and R.S. calculated the model interaction potential. All authors have contributed to writing of the manuscript with C.S. and G.K. as main contributors.

## Competing interests

The authors declare no competing interests.
