## [Peer Review File · Nature Communications]

REVIEWERS' COMMENTS:

Reviewer #1 (Remarks to the Author):

Overall recommendation: This is a very well-written and accessible communication paper on a topic of interest to the MOF community, and more broadly to those working in chemistry of materials and/or computational materials prediction. Subject to minor revisions, it would not be out of place in Nature Communications.

Key results: The authors report an easy to use and low-cost* screening framework to guide the selection of nitrile-based crosslinkers for incorporation into MOFs. Out-of-the-box, the code provided can be used for a wide range of Cu-based MOF systems and a library of 20 cross-linkers. The code provides experimentally-useful results within minutes. The screening-experimental workflow has been demonstrated for HKUST-1, resulting in the synthesis of novel retrofitted MOF materials.

* - the computational cost of the software is not quantified in the manuscript/SI. I report rough timings from testing the supplied script.

Validity: There are no fundamental flaws apparent in the manuscript or supplied code. The SI must be updated to include details of the N₂ adsorption measurements and BET characterisation (i.e. the pressure range over which BET was determined; it is good practice to include the linearised BET plot, with pressure range and calculated value of C indicated, for all samples).

Originality and significance: The screening approach developed and reported in this work is original, as are the majority of the reported retrofitted MOFs. Some computational studies on specific cross-linked (retrofitted) MOF systems have been previously reported, as has the TCNE-retrofitted HKUST-1. Published data inform the present work and are cited appropriately. The retrofitting of MOF systems is of general interest to the MOF community and rapid in silico screening of viable candidate cross-linkers is likely to be valuable to a relatively large number of experimental groups. As noted by the authors, the method described herein could be readily extended to some COFs, but also to metal-organic polyhedra. Importantly, the authors have included the code they have developed for dissemination – the true potential of the reported work is in its uptake by the MOF community as a means to accelerate materials development.

Data & methodology: The validity of the reported computational data is dependent on three primary factors: (i) DFT minimisation of CL molecules; (ii) determination of the MIP via DFT using a simple test system; (iii) accuracy of the MIP interpolation and resulting energy barrier. The level of theory employed in DFT calculations for (i) and (ii) is appropriate and has been employed for guest-MOF systems previously, including Cu-based systems. The test system used in (ii) introduces a limit to accuracy – the RetroFit code is optimising the N-Cu separation without considering interaction between the CL and the MOF more broadly. Given the chemical similarity of the CLs tested and the goal of a rapid screening (i.e. ranking of CLs, rather than high-level structure prediction), this is an acceptable simplification in my opinion. Only CL 19 is likely to be less well described than the others, with potentially close contact between the heterocycle N and the MOF backbone.

(iii) is dependent on user input (i.e. how many steps are included in the testlist in RetroFit.config), and on the resolution of the generated MIP (in this work 0.2 Å and 5°). Both resolutions are likely to be appropriate, but the uncertainty in calculated energy penalties and optimal positions introduced from this is not quantified in the manuscript or SI. The authors correctly note in SI that the interpolation should only be undertaken in the limits of the initial MIP.

The computational results reported in the manuscript were successfully reproduced by the reviewer using the supplementary code and MIP/CL data provided for HKUST-1 (tested under Python 3.6 on a Unix build on x86 architecture).

The experimental synthesis and characterisation methods used and their interpretation appears to be appropriate (see note below re: reviewer expertise). N₂ adsorption and BET measurements

have not been described in the manuscript or SI – this information must be included for the manuscript to be published.

Appropriate use of statistics and treatment of uncertainties: In general, the authors consider and report the limitations of the methods employed, which in the context of development of a rapid screening tool is appropriate. As noted above, the resolution of the MIP introduces an uncertainty in the predicted optimal positions and energy barriers; I am unsure what this uncertainty is likely to be. Can the authors comment on this or provide a numerical estimate for the uncertainty (key point: the authors report ΔE to 0.01 kcal mol⁻¹ – I'd like them to convince me that is justified)?

Conclusions: Subject to the point on uncertainty raised above, I have no concerns regarding the conclusions reported – they are appropriate and justified based on the data provided. Limitations of the method and future work are clearly stated. For future work, it should be possible to readily include pore-limiting diameter (compared to geometric information of the CL) as a crude diffusion/accessibility criterion in the screening process; several open-source PLD tools are available.

References: No concerns regarding referencing – sources are used and cited appropriately throughout.

Clarity and context: I congratulate the authors on their writing - the manuscript and SI is easy to follow, accessible and extremely well-written throughout, with minor typographical corrections. Appropriate context is given. I suggest the authors ask a critical friend (from a non-computational background) to try follow the How-To instructions provided and provide feedback. The code worked out-of-the-box for me, but I have been working with Linux for years and have access to HPC facilities with Python, ASE and other required libraries pre-configured.

Suggested improvements

Key revisions:

- The supplied code or How-To does not contain a licensing statement or information on how to acknowledge the authors – I strongly encourage the authors to consider how they would like the code to be used, modified and cited and include an appropriate (CC?) license.
- Inclusion of N₂ adsorption experimental details in SI, including BET plots and information.
- Estimate of uncertainty in the calculated energy barriers.
- Inclusion of benchmark software timings in SI. I suggest including the timings for the script as-supplied, as well as order-of-magnitude estimates for the MIP calculation and CL minimisation (although clearly these steps depend on the software, hardware and level of theory employed by the user), to further inform groups interested in extending RetroFit.

Typographical corrections and manuscript clarifications:

- Line 203: 'suitable CLs candidates' → 'suitable CL candidates'
- Line 221: 'dinitrle' → 'dinitrile'
- Line 244: include CL reference number to the statement regarding 3- and 4- membered rings
- Suggest improving the contrast between the N atoms and blue background in Figures 7 and 8 (perhaps outlining the atom would work).
- Line 358: Reference should be to Figure 8.
- In the SI, Table S1 and Eqn 1 should be Cu1-Cu-2-Cu4 angle (currently 1-2-3)

Improvements to RetroFit script and How-To:

- RetroFit sMOL output currently reports data to full precision – I suggest this output is truncated to reflect the uncertainty in calculated values as reported in the manuscript, or a note is added somewhere to suggest a truncation to users.
- You may also consider sMOL reporting in rank ordered format (e.g. based on energy penalty), for ease of use.
- Variable names in RetroFit.config do not always reference the equivalent variable names in the manuscript – I suggest the How-To is updated to explicitly connect MMM_angle to the Cu1-Cu2-

Cu4 angle referred to in-text.

- The authors should consider how the script and associated MIP/CL libraries would be best distributed in future, taking into account ease of including updates and bug-fixes, and recording download/usage data for future funding proposals, etc. Supplementary data with the paper may be less effective in the long-run than linking to a GitHub or other live repository.
- The authors should consider enlisting a critical, non-computational friend to work through the How-To instructions. I also suggest clarifying the OS(s) with which the supplied code is compatible.
- Further proof-reading of the How-To is required.

Reviewer Expertise:

My expertise is in the computational study of guest-MOF interactions, adsorption processes, MOF characterisation, and high-throughput screening of MOFs for adsorption processes. This includes both classical and quantum chemical simulation methods. I collaborate extensively with experimental colleagues in MOF synthesis, characterisation, and adsorption, so while I have broadly assessed the general appropriateness and interpretation of the experimental methods employed, I am not an expert.

Reviewer #2 (Remarks to the Author):

This is a nice, solid, and technically correct work. The techniques and concepts used here are adaptations of previous works. The results of "retrofitting" (basically another terminology for postsynthetically modified MOFs ?) are good and validate the computational method. This is not surprising. The authors recognize the importance of diffusion limitations in moving linkers in (or out) of MOF materials (e.g., JACS, 139 (16), 5906-5915, 2017). Some of these retrofitted materials could end up being core-shell structures.

While the idea of fitting new linkers into open metal/undercoordinated sites is conceptually valid and straightforward to understand, the authors may also mention work on inserting linkers into fully coordinated structures like ZIFs. They mention SALE techniques, but there are other ways of doing this. E.g., use of controlled acid gas treatments to partially demolish a ZIF and then insert a fresh linker. This lowers the thermodynamic barrier to insert new linkers or even to 'heal' the material with fresh linkers of the same type (see, for example, ACS Appl. Mater. Interfaces, 2017, 9, 40, 34597-34602).

I am OK with publishing the work in this journal or a more specialized one....it is up to the editors.

Reviewer #3 (Remarks to the Author):

In "Retrofitting Metal-Organic Frameworks" the authors propose a simple yet effective method for assessing the possibility of using a ditopic molecule as a cross linker (CL) between two open metal sites (OMS) in a Metal-Organic Framework (MOF). The method has the advantage to enable fast screening of CL/MOF pairs with reasonable accuracy. The potential of retrofitting as a post-synthetic modification method is well highlighted in the introduction of the manuscript. The authors also provide an open source code that aims to be used by experimentalists who would be interested in screening potentially interesting materials for retrofitting before going to the lab. To the best of my belief, no other open source tool is currently available to perform this particular task, which makes the outcome of this work both useful and original. Since this is a quite unexplored field, this tool could potentially help the field move forward. I do believe however that further work needs to be done to guarantee that the average experimentalist will be able to use the tool.

Here are my comments/questions to the authors:

(1) The Model Interaction Potential (MIP) is one of the inputs of the code. If I understood correctly, to get the MIP one must follow these steps:

- (i) choose a suitable model system for the single point DFT calculations, and
- (ii) perform such DFT calculations.

Do the authors intend for the user of RetroFit to perform these steps or do they intend to provide the MIP directly themselves?

If they intend for the user to do them, I have my concerns on whether these tasks can be performed by an experimentalist. To address this: could some guidelines for (i) and (ii) be included in the How-To documentation? Regarding (ii), what would happen if the calculations were done with a poor choice of parameters (for example, cutoff for pseudos not converged, poor choice of functional...)? Can the authors comment on the impact they expect this to have on the results?

If, on the contrary and as hinted in the conclusions, the authors are not expecting the users to compute the MIP, but are instead planning on creating a library with these data themselves; then I think that this is important information for the user, and that it should be mentioned earlier in the manuscript (rather than only in the conclusions). Authors should state that even though the method is universally applicable to all ditopic CLs/MOF pairs, the current version of the library only allows to study the Cu-paddlewheel OMS MOFs with the 20 CLs shown in Figure 3, and that this library will be extended in the (near) future to allow for more cases of use.

(2) All throughout the manuscript and supporting information, DeltaE values in kcal/mol are provided with a two to three decimals precision. What is the error of the energies from the DFT calculations? Does it make sense to use three decimals in the values as done for example in Table S3?

(3) If I understood correctly, once the code has been applied, a list of DeltaE values is obtained as the output, and then the user needs to decide whether these DeltaE are "low, medium or large" to see if it is worth pursuing the synthesis for each particular binary system (this distinction between "low, medium or large" seems to be important for arriving to a conclusion as per what reads in page 8 of the manuscript lines 264-268). Can the authors propose a concrete way of defining "low, medium and large" DeltaE values? I understand this in the context of a screening process, where data can be easily grouped (for example, one could plot the distribution of data and see whether it is unimodal, bimodal, etc, or think a criterion such as that if the DeltaE value lies in the first quartile it is "low", and from the fourth quartile onwards the value is "large"). But how can a user who is only interested in a few CL/MOF pairs and wants to know whether these are synthesizable or not, do this?

(4) Figure 2: the geometric parameters are very well explained in the supporting information but from what is currently written in the main manuscript I don't get how gamma is defined: if we note an angle by three points in space ABC (B being the vertex of the angle), I understand from Figure 2 that A and C are located on the two metal centers in the definition of gamma, but the location for B is unclear. Idem for alpha. To avoid confusion, I suggest that either the description of these parameters be clarified in the main manuscript, or just mention that they will be explained in the supporting information.

(5) Page 10 of the SI lists the current limitations of RetroFit. I believe this is important information for the users to know before trying to apply the code to their particular case of study, so I think this information should be included in the main manuscript.

(6) In general, I think that the code and documentation as provided, are in excellent shape to be used by a modeler or by an experimentalist with some experience in modeling and in using

python. I understand that this is version 1.0 of RetroFit, and that the authors naturally wish to publish the method together with a first version of the code, so this comment may be taken as something to be done in the future, to improve usability for an average experimentalist. Another option would be to change the target user, for instance: (a) to target a modeler as user who could run the code for the experimentalist, or (b) to state that some prior knowledge is required from the experimentalist user. In general, I think that if the target user is an average experimentalist, the code should be provided either as a stand-alone software, or as a plugin of a suite already used by experimentalists (such as Materials Studio for instance) or as a web application. Installing ASE and running a python code might be quite challenging for the average experimentalist if they don't have some prior python experience. Same for running OpenBabel. In addition, the How-To file should be double checked to make sure that the vocabulary used therein is understandable to the target public (it is currently crystal clear for a modeler, but I am not so sure that it would be clear for an experimentalist, for instance many of them do not know what an "xyz-file" is, nor what the "path to a directory" is).

Reviewer 1, suggested improvements

Key revisions:

- The supplied code or How-To does not contain a licensing statement or information on how to acknowledge the authors – I strongly encourage the authors to consider how they would like the code to be used, modified and cited and include an appropriate (CC?) license.

The python code is now available on GitHub (<https://github.com/GKieslich/RetroFit>) with an MIT license.

- Inclusion of N₂ adsorption experimental details in SI, including BET plots and information. Experimental details for N₂ adsorption experiments as well as calculation of the BET surface area has been added to the 'Methods' section in the main manuscript.

- Estimate of uncertainty in the calculated energy barriers.

A detailed discussion on the calculation and evaluation of the errors originating from the interpolation of the MIP has been added the supplementary information, see Supplementary Note 6. Additionally, a reference to the Supplementary Note 6 has been added to the text.

- Inclusion of benchmark software timings in SI. I suggest including the timings for the script as-supplied, as well as order-of-magnitude estimates for the MIP calculation and CL minimisation (although clearly these steps depend on the software, hardware and level of theory employed by the user), to further inform groups interested in extending RetroFit.

A measurement of the execution time was added to the code and is reported in Supplementary Note 3.

Typographical corrections and manuscript clarifications:

- Line 203: 'suitable CLs candidates' → 'suitable CL candidates' *This has been corrected.*
- Line 221: 'dinitrle' → 'dinitrile' *This has been corrected.*
- Line 244: include CL reference number to the statement regarding 3- and 4- membered rings. *This has been added in the revised manuscript.*
- Suggest improving the contrast between the N atoms and blue background in Figures 7 and 8 (perhaps outlining the atom would work).

- Line 358: Reference should be to Figure 8. *This has been corrected.*
- In the SI, Table S1 and Eqn 1 should be Cu1-Cu2-Cu4 angle (currently 1-2-3). *This has been corrected.*

Improvements to RetroFit script and How-To:

- RetroFit sMOL output currently reports data to full precision – I suggest this output is truncated to reflect the uncertainty in calculated values as reported in the manuscript, or a note is added somewhere to suggest a truncation to users.

We have added a comment into Supplementary Note 6 that based on our error calculations a truncation is suggested to three decimal places, i.e. 1.234.

- You may also consider sMOL reporting in rank ordered format (e.g. based on energy penalty), for ease of use.

This has been changed in the updated version of the code.

- Variable names in RetroFit.config do not always reference the equivalent variable names in the manuscript – I suggest the How-To is updated to explicitly connect MMM_angle to the Cu1-Cu2-Cu4 angle referred to in-text.

We thank the reviewer for this comment. We intend to keep the variable names as general as possible so that the code can easily be applied for other systems. In the revised manuscript we have changed this where appropriate.

- The authors should consider how the script and associated MIP/CL libraries would be best distributed in future, taking into account ease of including updates and bug-fixes, and recording download/usage data for future funding proposals, etc. Supplementary data with the paper may be less effective in the long-run than linking to a GitHub or other live repository.

We would like to thank the referee for raising this point. We have created a GitHub repository (<https://github.com/GKieslich/RetroFit/>) where the latest version of the program can be found. Furthermore, we have added a comment to the Methods Section referencing to the GitHub repository.

- The authors should consider enlisting a critical, non-computational friend to work through the How-To instructions. I also suggest clarifying the OS(s) with which the supplied code is compatible.

This has been done.

- Further proof-reading of the How-To is required.

This has been done.

Reviewer #2 (Remarks to the Author):

While the idea of fitting new linkers into open metal/undercoordinated sites is conceptually valid and straightforward to understand, the authors may also mention work on inserting linkers into fully coordinated structures like ZIFs. They mention SALE techniques, but there are other ways of doing this. E.g., use of controlled acid gas treatments to partially demolish a ZIF and then insert a fresh linker. This lowers the thermodynamic barrier to insert new linkers or even to 'heal' the material with fresh linkers of the same type (see, for example, ACS Appl. Mater. Interfaces, 2017, 9, 40, 34597-34602).

We added a reference in the main manuscript: "Moreover, it has been shown that the properties of acid-gas degraded zeolitic imidazolate frameworks can be recovered by post-synthetic treatment with a fresh linker solution."^[20]

Reviewer #3 (Remarks to the Author):

Here are my comments/questions to the authors:

(1) The Model Interaction Potential (MIP) is one of the inputs of the code. If I understood correctly, to get the MIP one must follow these steps:

- (i) choose a suitable model system for the single point DFT calculations, and
- (ii) perform such DFT calculations.

Do the authors intend for the user of RetroFit to perform these steps or do they intend to provide the MIP directly themselves?

If they intend for the user to do them, I have my concerns on whether these tasks can be performed by an experimentalist. To address this: could some guidelines for (i) and (ii) be included in the How-To documentation? Regarding (ii), what would happen if the calculations were done with a poor choice of parameters (for example, cutoff for pseudos not converged, poor choice of functional...)? Can the authors comment on the impact they expect this to have on the results?

If, on the contrary and as hinted in the conclusions, the authors are not expecting the users to compute the MIP, but are instead planning on creating a library with these data themselves; then I think that this is important information for the user, and that it should be mentioned earlier in the manuscript (rather than only in the conclusions). Authors should state that even though the method is universally applicable to all ditopic CLs/MOF pairs, the current version of the library only allows to study the Cu-paddlewheel OMS MOFs with the 20 CLs shown in Figure 3, and that this library will be extended in the (near) future to allow for more cases of use.

We thank Reviewer #3 for raising this question. The steps to obtain a MIP are correctly understood by the reviewer and he is right in pointing out that the generation of new MIPs will be challenging for the average user of RetroFit. In fact, our intention is that advanced users and computational chemists develop RetroFit and provide new MIPs while experimentalists can use RetroFit as a guideline for retrofitting experiments. We clarified this in the methods section of the revised manuscript.

(2) All throughout the manuscript and supporting information, DeltaE values in kcal/mol are provided with a two to three decimals precision. What is the error of the energies from the DFT calculations? Does it make sense to use three decimals in the values as done for example in Table S3?

The calculation and evaluation of the errors originating from the interpolation of the MIP are extensively discussed in Supplementary Note 6.

(3) If I understood correctly, once the code has been applied, a list of DeltaE values is obtained as the output, and then the user needs to decide whether these DeltaE are “low, medium or large” to see if it is worth pursuing the synthesis for each particular binary system (this distinction between “low, medium or large” seems to be important for arriving to a conclusion as per what reads in page 8 of the manuscript lines 264-268). Can the authors propose a concrete way of defining “low, medium and large” DeltaE values? I understand this in the context of a screening process, where data can be easily grouped (for example, one could plot the distribution of data and see whether it is unimodal, bimodal, etc, or think a criterion such as that if the DeltaE value lies in the first quartile it is “low”, and from the fourth quartile onwards the value is “large”). But how can a user who is only interested in a few

CL/MOF pairs and wants to know whether these are synthesizable or not, do this?

We thank the reviewer for mentioning this. We added a detailed discussion about the categorization of the energy penalties as output values from RetroFit to Supplementary Note 7 and referenced this discussion in the main manuscript.

(4) Figure 2: the geometric parameters are very well explained in the supporting information but from what is currently written in the main manuscript I don't get how gamma is defined: if we note an angle by three points in space ABC (B being the vertex of the angle), I understand from Figure 2 that A and C are located on the two metal centers in the definition of gamma, but the location for B is unclear. Idem for alpha. To avoid confusion, I suggest that either the description of these parameters be clarified in the main manuscript, or just mention that they will be explained in the supporting information.

The author is right in his argumentation. Of course, an angle is always defined by three points ABC. In the case of a Cu paddlewheel MOF, the definition of gamma is straight forward, as the Cu-Cu axes defines the vector between M and its OMS. This vector is indicated in Figure 2. When applying RetroFit to new MOF systems, the user must define the M-OMS vector. A short discussion is added in Supplementary Note 3.

(5) Page 10 of the SI lists the current limitations of RetroFit. I believe this is important information for the users to know before trying to apply the code to their particular case of study, so I think this information should be included in the main manuscript.

We agree with the referee that it is important to recognize the limitations of the model, which are discussed several times in the manuscript, e.g. "The largest deviations occur due to CL deformations, which are not accounted for in our model.". Given the importance to acknowledge the limitation of the system, we added the list in the SI as an overview (see Supplementary Note 4); however, we believe that this issue is sufficiently addressed in the current version of the manuscript.

(6) In general, I think that the code and documentation as provided, are in excellent shape to be used by a modeler or by an experimentalist with some experience in modeling and in using python. I understand that this is version 1.0 of RetroFit, and that the authors naturally wish to publish the method together with a first version of the code, so this comment may be taken as something to be done in the future, to improve usability for an average experimentalist. Another option would be to change the target user, for instance: (a) to target a modeler as user who could run the code for the experimentalist, or (b) to state that some prior knowledge is required from the experimentalist user. In general, I think that if the target user is an average experimentalist, the code should be provided either as a stand-alone software, or as a plugin of a suite already used by experimentalists (such as Materials Studio for instance) or as a web application. Installing ASE and running a python code might be quite challenging for the average experimentalist if they don't have some prior python experience. Same for running OpenBabel. In addition, the How-To file should be double checked to make sure that the vocabulary used therein is understandable to the target public (it is currently crystal clear for a modeler, but I am not so sure that it would be clear for an experimentalist, for instance many of them do not know what an "xyz-file" is, nor what the "path to a directory" is).

We thank the referee for the very good advice for future developments of the program. In fact we are aiming to make future versions of the program more user-friendly to make it more experimentalists friendly.